# Animals, Sages and Saints: Alfonso Vagnone's Rhetorical Strategy in Chinese Context

**Sha Liu**

Department of Religious Studies, Faculty of Philosophy, Fudan University, Shanghai 200437, China;
21110160039@m.fudan.edu.cn

**Abstract:** Around 1632, while stationed in Shanxi Province (China), the Italian Jesuit Alfonso Vagnone published a pedagogical treatise entitled *On the Education of Children* (*Tongyou jiaoyu*) on which he had worked for several years. The book achieves a carefully crafted synthesis between Confucian educational principles, European Humanism, and Jesuit pedagogy. This is achieved through various rhetorical devices, one of them being the extensive use of animal simile, which prepares considerations about behavioral models to be found in (Pagan) Sages and (Christian) saints. This study focuses on the rhetorical and narrative methods through which Vagnone grounds a gradation and continuum between Nature and Grace, inserting his pedagogical considerations into a carefully crafted apologetics. Vagnone's work is also remarkable by its implicit openness to various Confucian schools, while Matteo Ricci was drawing a sharp distinction between original Confucianism and the School of the Principle. In more than one way, Vagnone's work is "ecumenical". It makes the love for one's offspring, found in all living species and in all nations, the ground of moral reform, itself conducive to greater openness toward revealed truths.

**Keywords:** animal stories; Confucianism; education; European Humanism; Jesuits in China; parables; pedagogy; School of the Principle; rhetoric

## 1. Introduction

Around 1632, the Italian Jesuit Alfonso Vagnone (Gao Yizhi, 高一志, 1568–1640) published a pedagogical treatise entitled *On the education of Children* (*Tongyou jiaoyu* 童幼教育).[1] In his preface to the book, the well-known scholar Han Lin ( 韩霖, 1559–？)[2] underscores the crucial role that the care brought to the education of young children (*mengqiu* 蒙求)[3] needs to play when striving to implement the Confucian vision of a world of "great unity" (*datong* 大同).[4] Han Lin applauds the contributions made in the field by the *Dadai Liji* ( 大戴礼记, *Records of ritual matters by Dai the Elder*) and *Dizizhi* ( 弟子职, *Duties of the student*), which are two books dated from the Western Han dynasty. He also decries the neglect for *mengqiu* that was shown at other times. He notably laments the Qin Dynasty's adherence to Legalism (*fajia* 法家), the Eastern Han Dynasty's preference for the Huang Lao 黄老 school of Daoism, the Jin Dynasty's reliance on "lofty discourses (*qingtan* 清談)" (a distinctive feature of "Arcane studies" (*xuanxue* 玄学)), and the Tang Dynasty's passion for the *cifu* 词赋 literary form,[5] which led to the demise of the ancient approach to education. In addition, Han Lin criticizes the model prevalent at his time, arguing that the education provided by fathers and teachers (*fushi zhi jiao* 父師之教) focuses merely on the acquisition of official titles through the imperial examination system. In contrast, the primer authored by Zhu Xi 朱熹, *Xiaoxue* 小学, is portrayed by Han Lin as successfully responding to the deficiencies shown by the other modes of early childhood learning. Han Lin not only compares Vagnone's *Tongyoujiaoyu* with Zhu Xi's *Xiaoxue*, but he also stresses the fact that Vagnone's work, due to its excellent rhetorical and argumentative style, surpasses the latter: "The Master's words are in line with Zhu Xi's book, but [his] book is characterized by eloquent rhetoric (*hao ci* 好辭)" (Mei and Tan 2017, p. 150; Falato 2020, p. 137, modified).

This article takes as its starting point Han Lin's evaluation of Vagnone's *Tongyou jiaoyu* as a book characterized by "eloquent rhetoric". It analyzes the way Western thinking and Vagnone's unwavering Catholic faith have combined with the latter's "eloquent rhetoric" and knowledge of the Confucian tradition for creating a remarkably successful pedagogical primer. I pay special attention to three aspects of Vagnone's rhetoric: first, his use of animal analogies; second, the way these analogies are mobilized into a continuum that is both pedagogical and theological; third, his implicit openness to the whole of the Confucian tradition rather than merely to one particular school within it. By examining these particular features, I intend to highlight the fact that Vagnone's position as a Jesuit missionary located in China was shaped by three major ideological currents: the European Renaissance's Humanism; Counter-Reformation's theological inflexions; and the "return to the Classics" movement within Confucianism. Ultimately, this article aims to shed light on the relationship between Vagnone's Jesuit-inspired educational and theological model, on the one hand, and Confucian thought, both pre-Qin and Song-Ming, on the other hand.

## 2. Vagnone's Rhetorical Science

### 2.1. The Ciceronian Tradition

According to Vagnone, there is not much difference between humans and animals in the learning of survival skills, but it is the study of "literature" that truly distinguishes human beings from animals:

> In the West, from where I come, after the child has received initial education, he continues with the study of Letters. For it is skillful language that distinguishes humans from animals and makes him able to communicate with others, and thus at the start of study one cannot neglect to exercise in Letters. (Falato 2020, p. 239, modified)

> 吾西] 小童開蒙之後遂習于文。盖言者，人所以別于獸而交接于物，則始學無不宜修文者。 (Mei and Tan 2017, pp. 216–17)

In the fifth chapter of the second part of the work (*Western Studies* (*xixue* 西學)), Vagnone introduces Cicero's five-step Art of Rhetoric, which includes the choice of a subject (*inventio*), style (*elocutio*), structure (*compositio*), memory (*memoria*), and delivery (*actio*). Building on Cicero's rhetorical theory, the Jesuit scholar Cypriano Suarez had published in 1569 *De arte rhetorica*, which was later used by Jesuit schools as the basis for literary education. The description of Cicero's division of the rhetorical art comes immediately after the part just quoted:

> The Western Art of Letters can be roughly summed up in five aspects: firstly, one must investigate the situation of people and things in order to understand the underlying principles that should be spoken about, thus revealing their beautiful meaning. Secondly, one must pay attention to the orderly arrangement of ideas, the way a wise leader orders his troops, placing the brave at the forefront and the cowardly in the middle. Thirdly, one should use elegant and refined language to embellish one's ideas. Fourthly, one must refine and practice one's opinions, reciting them until they are fully internalized. Finally, what is hidden in our minds must be presented in a public forum, or in front of a group of wise scholars. When it comes to debate and argue, among these five aspects, the most important is to prioritize practical principles and put them into application. How could someone possess these language skills and allow one's words to be scattered in the wind? (Falato 2020, pp. 239–40, modified)

> 然太西之文，約歸於五而成，先究事物人時之勢而思具 所當言之道理，以發明其美意焉。次貴乎，先後布置有序,如師之智者。節制行伍。勇者置於軍之前後而懦者屯之於中。次以古語美言潤飾之。次以所成議論，嫻習成誦。黙識心胸终至于公堂或諸智者之前。辨誦之，此五者之中，必貴實理，而致于用焉，豈徒具其文而苟吐散于空中乎? (Mei and Tan 2017, p. 217)

By pointing out the central role of literary studies in the education of youngsters, Vagnone clearly privileges "nurture' over "nature":

> Ancient sages used to debate: "If one wishes to become outstanding, what is more important, nature or learning? It is certainly learning that needs to be considered more important. Good nature is like a fertile field, if it is not constantly managed it will surely sprout weeds. [Even] those who are considered stupid by nature will eventually succeed through study. Someone with a gifted nature succeeding without study is a sight that has never been seen." (Falato 2020, p. 229)

> 先智者，嘗設論曰: 欲成大器，質與學 孰重，則必以學爲重焉。夫美質者，如田之膏腴，非恆修治之, 徒長豐草 耳。嘗觀質之鈍者，多用學之功力，竟得成全。末見美質者， 不學而能成 也。是以古智以美質不學，譬之美軀之無兩目, 寰宇之無三光也。(Mei and Tan 2017, p. 210)

### 2.2. Accommodating the Naturalism of the School of Principle

Yet, Man is anchored into Nature, as shown by Vagnone's extensive use of animal analogies in the *Tongyou Jiaoyu*. The use of this rhetorical device is in harmony with a basic precept propounded by the School of Principle (*Lixue* 理學): "Using natural laws to clarify human affairs (*tui tiandao yi ming renshi* 推天道以明人事)"[6], a precept that departs from pre-Qin Confucianism: the pre-Qin Confucian tradition was rarely concerned with "naturalism" nor did it often elevate observations of animals to ethical principles. (It goes without saying that our discussion would be totally different if the Jesuits' main interlocutor had not been Confucianism but the doctrine originating from the *Zhuangzi* for instance—see (Lynn 2019).) In fact, Matteo Ricci (1552–1610) had implicitly rejected the naturalistic tendency of the School of Principle and insisted upon the fact that that pre-Qin Confucianism was more conducive to the spread of the Catholic faith than latter versions of the same tradition.[7] In early Confucianism, the clear distinction to be made between humans and animals is emphasized in several passages of the *Xunzi* 荀子, and it is exemplified in the following passage from the *Mencius* 孟子:

> Once people have plenty of food and warm clothes, they lead idle lives. This is their Way. Then, unless they are taught, they are hardly different from the birds and animals. The sage emperor worried about this. H made Xie minister of education so the people would be taught about the bonds of human community; affection between father and son; duty between sovereign and subject; responsibility between husband and wife; proper station between young and old; trust between friend and friend. ("Ten Wen Gong" 1.4, transl. Hinton (1999, p. 92), modified)

> 人之有道也，飽食、煖衣、逸居而無教，則近於禽獸。聖人有憂之，使契為司徒，教以人倫：父子有親，君臣有義，夫婦有別，長幼有序，朋友有信。

For sure, early Confucian texts attribute human features to animals (the pheasant for instance being endowed with honesty and nobility) but they merely *suggest* such equivalences, often doing so in an ambiguous fashion, and such analogies will not be developed and systematized until Song-Ming scholars engage into the task (Wu 2020). If Vagnone's use of animal parables conforms to the Greek and Latin rhetorical tradition, in the Chinese context, this rhetorical strategy cannot be seen only as an embellishment meant at pleasing and convincing his audience (although this aspect should not be overlooked): it inscribes natural and human realities into the continuum that the School of Principles had been emphasizing.

What is more, Vagnone highlights lessons in educational wisdom that can be drawn from the animal realm:

> At birth bears are merely [a ball of] flesh and their cubs are not perceived [by their own kind] as being bears, therefore [the mothers] lick their young for correcting the matter, till they become like bears. [ . . . ] All animals want their young to

be like them, so should people not want their offspring to do the same? ([Falato 2020](), pp. 154–55, modified)

熊生僅肉團耳，自視之非熊也，常用舌餂摩脩治之，漸漸成熊矣 [。。] 禽獸皆
欲其子肖，人乃不欲其子肖耶? ([Mei and Tan 2017](), p. 163)

In the West, the use of animal parable to illustrate ethical principles is a well-established rhetorical device. This approach is often used in order to draw parallels between animal behavior and human nature as well as provide guidance for moral education. Vagnone cites examples coming from the animal kingdom, such as the magpie, the ostrich, the deer, the raccoon the dog, the wolf, the elephant and the bear to illustrate practices that foster sense of kinship, promote nursing, or determine the appropriate time to enter into marriage. On this last point, Vagnone states that those who are considering marriage should be able to wait until the moment has come: even animals who fly and run wait for the right season to mate, and they do not mate or breed outside this time. Therefore, humans should also be patient and wait for the appropriate time to start a family (see [Falato 2020](), p. 140; [Mei and Tan 2017](), p. 159).

These examples are generally drawn from the works of ancient authors, first and foremost from Pliny the Elder's *Historia naturalis* as well as from the writings of Erasmus and Renaissance zoologists, who all proclaim Nature to be the best teacher ([Mei and Tan 2017](), p. 108). Vagnone associates the Western humanistic tradition with the Chinese tradition, finding ethical and educational patterns in examples taken from the natural world. Vagnone operates a few cross-cultural adaptations, for instance when he attributes to the phoenix educational techniques ascribed by Pliny to the eagle or when he substitutes the name of the nightingale for a species common in China. (on these two examples, [Falato 2020](), pp. 112–13).

### 2.3. Animal Parables as an Emplotment Device

Should we speak of animal "analogies", "fables" or "parables"? The difference is sometimes hard to tell, and it becomes irrelevant when we simply define a parable as a "short stories with allegorical significance" ([Vermander 2022a](), p. 5, after [Meier 2016]()): even if animal references function as mere simile, they are always anchored into embryonic story-telling: the phoenix teaches its hatchlings to look at the sun without blinking, and it ostracizes the ones who cannot do so; likewise, the elephant teaches its youngsters to dance and purify themselves at each new moon ([Falato 2020](), p. 296); the nightingale, the deer, the fox and the wolf spare no effort to teach their offspring to respectively sing, run, charm and hunt, each species teaching its youngsters skills that derive from its nature but all sharing the same zeal for education ([Falato 2020](), pp. 266–67). The lesson taught by these fables is double: (a) one must act according to one's specific nature; (b) all species obey a law that fosters the love and care for one's offspring. In so far as human beings do not recognize such principles and do not act accordingly, they run the risk of becoming degenerated, i.e., to harm irremediably their very nature.

A corpus of interrelated parables can thus suggest an overall emplotment model, a "metanarrative [ . . . ] directed by the observation of some foundational realities" ([Vermander 2022a](), p. 4). The emplotment model privileged by Vagnone is straightforward: it corresponds to a model of gradation, which is often found in the Gospels and in biblical antecedents. "If God so clothes the grass of the field, which is alive today and tomorrow is thrown into the oven, will he not *much more* clothe you—you of little faith?" (Lk 12.28) "If you then, evil as you are, know how to give your children what is good, how *much more* will the heavenly Father give the Holy Spirit to those who ask him!'" (Lk 11.13) In the first example, the care shown by God towards the vegetal realm is felt as becoming necessarily even greater when the apex of creation—the human being—is concerned. In the second example, the imperfect care that people extend to their children announces the perfect love and gift bestowed by God upon them. Similarly, in Vagnone's logic: (a) the educational devotion displayed by inferior creatures must incite parents and teachers to bring to the fullest their pedagogical zeal and expertise; (b) as we will see, pedagogical excellence is

gradually displayed in the hierarchy that extends from animals to sages and then to saints, on the one hand, and from natural to human and finally divine knowledge, on the other hand.

## 3. Vagnone's Argumentative Strategy

### 3.1. Not Truly a "Renaissance Man"

Vagnone's extensive use of classical authors as well as Erasmus and scientists from the same period could easily make the reader identify him with a "Renaissance figure." This would constitute a slightly mistaken perspective. The way Vagnone progressively crafts his argument needs careful investigation.

Thierry Meynard has suggested that the target audience of *Tongyou jiaoyu* was not Christians, as the book does not use extensively Christian hagiographies but rather stories from ancient Greece and Rome (Mei and Tan 2017, p. 117). Although this prevalence cannot be disputed, a precise account of quotes and mentions shows a diversified picture: 31 of the names being cited belong to ancient Greece and Rome—Plato (15 occurrences), Aristotle (6), Pythagoras (6), Diogenes (5) or Alexander (5). However, there are also 17 names from the Christian tradition (Li 1998, p. 116). Vagnone carefully suggests a hierarchy within his sources; he refers to ancient Greek and Roman figures as "literati" (*shi* 士) and "sages," (*xian* 賢) while speaking of Church saints as "saints" (*sheng* 聖). Vagnone also speaks of God as the "Creator" (*zaohuazhu* 造化主) and "Heavenly Lord" (*Tianzhu* 天主), which is the divine name that the Jesuits decided to adopt as a consequence of the Terms Controversy (1616–1633) that divided the China mission shortly after the death of Matteo Ricci (the controversy was bearing upon the Chinese expressions—or transliterations—to be adopted when translating the basic lexicon of the Christian faith). On the part of Vagnone, some of the lexical choices he made in the book were probably an act of obedience: at the time he was writing the *Tongyou jiaoyu*, he was still requesting that the translation of "God" as *shangdi* 上帝 be allowed (Ricci had used both *tianzhu* and *shangdi*), and he had written (in 1628) to the General of the Society of Jesus, protesting against the decision that allowed only for the term *tianzhu* to be used (Vu Thanh 2019, p. 415).

The lexical hierarchy that Vagnone develops in his book brings to mind the famous words of Zhou Dunyi 周敦頤 (1017–1073), the founder of the Song-Ming School of Confucianism:

> The saints emulate [or: admire] Heaven, the sages emulate the saints, and the literati emulate the sages. (*Tongshu*, *Zhixue*)
>
> 聖希天，賢希聖，士希賢。《通書。志學》

In his lexical choices, Vagnone combines Confucian traditions rephrased by the School of the Principle with an approach to Christian truths based on a gradualism that evokes Saint Ignatius' famous adage according to which, in spiritual conversations, "we enter his door with [our interlocutor], but we come out our own" (Loyola n.d.). Greek thinkers such as Diogenes, Plato, Pythagoras, Demosthenes, Pythagoras, Solon, Zeno or yet Phocion are mentioned with praise, notably for the way they used to attribute a child's defects to the shortcomings of parents or teachers (*Tongyou jiaoyu*, chapter 6), and they are all described as being noble personalities. Still, biblical figures and church saints rank higher than the figures from ancient Greece and Rome. For example, in various places, Vagnone refers to Plato and Pythagoras as "great sages of ancient times [上古大賢], or "masters of ancient times" [上古之學宗]" while Saint Jerome is referred to as "a Saint from the West [西之聖人] and Saint Malachy is referred to as "an ancient saint from a northern Western country [西北國古聖]". Likewise, David is a "holy ruler" (*shengzhu* 聖主).

There are only two exceptions to the above-mentioned pattern, which reveal a clever strategy on the part of Vagnone; when he advocates for the study of classics, he mentions two famous figures from the Church: Saint Ignatius of Loyola, the founder of the Jesuits, and Augustine of Hippo, who are both referred to as literati (*shi* 士) rather than sages (*xian* 賢) or saints (*sheng* 聖) (Falato 2020, pp. 235–36; Mei and Tan 2017, p. 214). This uncon-

ventional reference to these two Church figures is not a result of Vagnone's negligence but rather a deliberate emphasis on the misguided paths that both figures took before converting to Catholicism: Saint Ignatius was initially "fond of bravery" and had not yet mastered practical learning, but he became a "teacher of enlightened thought" after reading books on the lives of the saints following his injury on the battlefield. Augustine of Hippo, on the other hand, "unfortunately studied under heretical teachers in his youth and was lost to the heretical books, but his heart was never at peace". It was only after reading Paul's epistles that he "abandoned heresy and entered the holy realm". Thus, Vagnone links together two famous conversion stories by insisting on their common thread; it is by reading virtuous books that both men entered the right path.

Vagnone makes his first direct reference to the Bible when emphasizing the importance of fearing reproach and humiliation as the foundation of all learning. Further explicit biblical references insist upon the stress that the Bible puts on filial piety—an obvious accommodation strategy. In the "Western Learning" section of the second volume of the book, when discussing the Jesuit educational program known as *Ratio Studiorum*, Vagnone emphasizes the following:

> Above human studies there are heavenly studies that are called *theologia* in the West. These studies are based on the ancient and modern classics, along with texts written by saints and sages. [Theology] analyses the origin of the righteous doctrine, refuting the evils of heterodoxy. This learning is also divided into four great branches: the first states that above all things there must be a Lord who is the greatest, brightest, kindest, and fairest. His nature is all-encompassing and His qualities are wondrous. (Falato 2020, p. 247)

> 人學之上尚有天學。 西土所謂陡羅日亚也。 此學乃依古今經典， 與諸聖賢註論。 剖析正道之本源， 而攻闢異端之邪也。 其 學亦分四大支。 一論物上必有一主， 至大至明至善至公。 即詳其性， 及其妙情。 (Mei and Tan 2017, p. 220)

At first look, Vagnone's outlook is still anchored into scholasticism, more theo-centric than anthropocentric. The focus on learning and the rightful doctrine reminds the reader of Niccolo Longobardi's stress on orthodoxy[8] rather than Ricci's encompassing natural theology (see Canaris 2021) without the former totally excluding the latter. Additionally, the transliteration of the term "theology" ( 陡羅日亚) is reminiscent of the caution shown by the Jesuit missionaries formerly stationed in Japan, who had started the so-called Terms Controversy: Influenced by Francis Xavier's way of preaching, the Society of Jesus in Japan had made theological and linguistic choices that were far less accommodating than those of Ricci, and he had transliterated the name of God from the Latin Deus. The end of the sixteenth century and the beginning of the following century were marked by a theology of a more pessimistic outlook than the training that Ricci had enjoyed during his formative years. After Ricci had died, almost all the Jesuits in the province of Japan and some of the Jesuits in the vice-province of China protested, often violently, against the linguistic and ritual accommodations already made, and they even threatened to appeal to the Inquisition based in Manila. The majority of the Jesuits of the China mission, led by Nicolas Trigault (1577–1628) and Alfonso Vagnone, were remaining faithful to the accommodations decided by the founder of the mission (Duan 2017; Falato 2020, pp. 57–58). After the conference of Jiading (near Shanghai) which, starting in December 1627, brought together eleven of the Jesuits involved in the controversy, the Visitor, André Palmeiro (1569–1635) decided to authorize the use of the term Tianzhu (Lord of Heaven) to designate the Christian God but to banish those of Shangdi (Lord of Above) and Tian (Heaven), which Ricci was also using. On the other hand, the lawfulness of attending rites in honor of Confucius and the ancestors was confirmed (see Pina 2003; Kim 2004; Brockney 2014). When reading Vagnone's presentation of what *theologia* is meant to be, it first looks as if finding a Chinese equivalent name of this sacred science would have run the risk of corrupting the purity of the doctrine it conveys.[9] However, as we have seen, during the term's controversy, Vagnone was pleading for preserving Ricci's lexical openness against the strictness shown by Longobardi and

Jesuits having arrived from Japan, as persecutions destroyed this formerly flourishing mission. Some lexical choices of the *Tongyou jiaoyu* may have less to do with Vagnone's own choices than with the debate raging at that time and the necessity of securing approval from his superiors before publishing the book: the printed edition informs us that the text was revised by the Jesuits Gaspar Ferreira, Niccolo Longobardi and Johann Terrenz Shreck before it was approved by the vice-provincial Manuel Dias Jr. "Revisions" may imply a mere examination that will focus on doctrinal orthodoxy, or it may entail modifications, be they lexical or content-related. In the acrimonious climate that was prevailing at that time within the Jesuit Chinese mission (the Terms Controversy was not fully concluded at that time), the latter hypothesis enjoys a high degree of probability.

### 3.2. A Theology of Nature and Grace

As far as his argumentative strategy makes us able to approach it, Vagnone's theology remains strongly inscribed in the Catholic mainstream. The use of animal parables in the Christian context originated with Origen and was adopted by Augustine (for a general history of Christian parables, see Gowler 2017). More generally, reliance on allegories highlights the Catholic Church's belief that between "the Book of Nature" and "the Book of Grace", a continuity can be drawn. Taken from the natural world, the simile that Vagnone makes use of constitutes proof enough that he fully adheres to the Catholic doctrine of "inheritance of natural grace" against the Protestant outlook focused on the "disruption of natural grace". The multiple meanings generated from allegories develop according to the way the reader envisions human nature (a parable is an open-ended story in the sense that its ultimate meaning is affixed by the reader). At the same time, the field of interpretations is not fully open: Vagnone's combined use of animal micro-stories and biblical references first and foremost testifies to the potential for good inscribed into human nature, even after the Fall, provided that the same nature be adequately helped and guided along the years of apprenticeship.

One may go as far as to contend that Vagnone's reliance on animal parables and similar techniques inscribes the worldview he shares with his readers into Catholic symbolism: the use of physical objects bestows visible forms to spiritual concepts, enabling the listener or the viewer to better understand and experience matters related to faith. Going yet one step further, the frequent references to biblical and Church figures highlights Vagnone's commitment to the Catholic sacramental tradition.

One discerns here a specificity of Vagnone's approach when compared to Ricci's *True Meaning of the Lord of Heaven* (*Tianzhu shiyi* 天主實義): in the case of Vagnone, the stress on the inheritance of natural grace serves as a bridge for entering into a (cautious) dialogue with the naturalism proper to Song-Ming Confucianism, while the distrust shown by Ricci toward the School of the Principle is well-known. Conversely, Ricci is at ease with the "natural theology" of pre-Qin Confucianism, more or less equated to Theism, while Vagnone, even if he refers to (scholastic) *theologia* when it comes to things divine, sees in the natural knowledge displayed by latter-day Confucians a bridge that leads to orthodox thinking but that stops at its sacred doors.

## 4. Assessing Educational Models

### 4.1. Ricci's Implicit Anthropology

Although the *Tianzhu shiyi* is generally seen—and justly so—as a book meant to foster positive and respectful dialogue, its assessment of Chinese practical and moral education is implicitly more negative than the one that Vagnone's treatise seems to suggest. In Chapter VII of the *Tianzhu shiyi*, Ricci makes "the capacity to infer and reason (*neng tuilun li* zhe 能推論理者)" a defining human characteristics (he does not follow Mencius' approach that gives precedence to the "four seeds" giving rise to Humaneness, Duty, Ritual and Wisdom—see *Mencius* 2A.6) while refusing to assimilate Reason (*li* 理) with Nature (*xing* 性) as Zhu Xi does. Ricci gives preeminence to the human capacity to discern and freely decide, contrasting it with the fact that minerals and animals cannot enact "good"

or "bad" deeds (*Tianzhu shiyi*, ch. VII).[10] Here, Ricci's use of animal metaphor creates a divide rather than a continuum within the Creation. In addition, Ricci stresses the fact that studying (*xue* 學) is not about recovering the state of nature that was supposedly once lost (an idea present not only in the School of the Principle but already in Confucius and Mencius—see notably *Mencius* 6A.11) but rather to enter into a new level of consciousness and understanding, i.e., to acquire the capacity to discern, which is a capacity that Man is not naturally endowed with (par. 434–438). The stress on becoming able to discern and then decide accordingly is a distinctive feature of Jesuit spirituality and education, which resonates with (but also slightly diverges from) the Confucian tradition (see Vermander 2022b).

In the same Chapter VII of the *Tianzhu shiyi*, criticisms against a model of teaching that relies mainly on the study of ancient Classics rather than on cultivating the capacity to discern and make good use of one's freedom becomes at times more stringent:

> To study does not mean only to follow the words and actions of the Ancients as they have been related, but also to enlighten oneself, this by scrutinizing Heaven, Earth and all things, so as to deduce from them knowledge concerning human affairs. This is why it is said: The wise man does not fear lacking books. If there is no teacher to instruct me, Heaven, Earth and all things are my all-encompassing teacher and book. (*Tianzhu shiyi*, par. 446)[11]

> 夫學之謂，非但專效先覺行動語錄謂之學，亦有自己領悟之學；有視察天地萬物而推習人事之學。故曰智者不患乏書冊、無傳師，天地萬物盡我師、盡我券也。

More generally, Ricci stresses an approach to education that starts from the heart and not from the disciplining of one's appearance and the memorizing of Classics. Likewise, the *Tongyou jiaoyu* speaks of moral education before treating of the study of Letters, Classics and Rhetoric. Still, some passages of the same book resonate in a way slightly different from the one produced by the *Tianzhu shiyi*:

> In the past, in the Western countries, there was a sage called Zeno, who dedicated himself from an early age to orthodox learning. He asked the gods how he could complete his studies. The gods replied: "The method is to engage often with those who have passed away and be diligent." Zeno carefully considered the gods' meaning for a long while and understood that "passed away" referred to the people of the past who had already died. [...] Studying, but giving up the classics, would be like wishing to fly without wings, is this possible? (Falato 2020, p. 231, modified)

> 西國古之名賢，曰責諾者, 初志於正學，則問於神，何由而成其學乎，神曰: 多交於終者而勤法之可也。賢思察神意良久，知終者，已死之古人也。[...] 學而舍典籍, 猶舍毛羽而欲高飛，豈可得乎? (Mei and Tan 2017, p. 211)

### 4.2. Vagnone and Humanistic Education

On the basis of the preceding analysis, we can now elucidate Vagnone's position on educational matters as well as the rhetorical strategy he uses. Some specific features appear clearly throughout the *Tongyou jiaoyu*:

Vagnone focuses on "role models", the latter being found in ancient Greece and Rome as well as in Catholic saints (with the subtle lexical differentiation already noted). References to Chinese sages are muted but nevertheless present; the praise of the practice of keeping silent as a privileged way to listen and to learn evokes *Analects* 19.1; the stress on the sense of shame as essential for moral conduct is reminiscent of *Mencius* 2A.6 and 7A.7. More importantly, Vagnone refers explicitly and positively to the doctrine of the Five Relationships (*wu lun* 五倫) developed by the same Mencius. Still, when it comes to the Chinese tradition, *values* are emphasized much more than living examples are, while the Western moral and educational model is introduced mainly through *figures*, probably so as to show that the West was not lacking in models of virtue similar to (or greater than) the ones familiar to a Chinese audience.

The stress on the continuity existing between the animal and human kingdoms allows for a sense of universality anchored into Natural Law. The fact that all people are able to read the "Book of Nature" justifies the commonalities in values and practices found between the West and China. More broadly, the love of one's offspring common to all living species, which fosters the wish to see youngsters fulfill their potential and destiny, explains the supposedly universal agreement on educational principles that Vagnone seeks to highlight and foster. The large number of agriculture-related metaphors, especially when it comes to marriage and the raising of one's family, anchors the text even more into a rhetoric that implies the recognition by all of the natural principles that apply to every nation and domain of activity.

Additionally, the *Tongyou* jiaoyu subtly pleads for the development of a system of local education that would follow the Jesuit system. After having showcased the crucial role played by teachers in the formation of children (suggested to be even more decisive than the one played by parents)[12], Vagnone writes:

> In each of the Western countries, the local Society of Jesus has established colleges, where wise teachers are invited from all around to devote themselves to the education of the youngsters. There are scholars who, as soon as they terminate their studies, disregarding emoluments and personal glory, attempt to provide a meritorious service to God and to serve their country with the establishment of such schools and teachings: this is what the Society of Jesus does. (Falato 2020, p. 164, modified)[13]
>
> 吾西諸國，每都會，又立一大學，遍邀賢師，專務幼教。又有士既成學，而不計罰祿，不計榮達，但圖立功於天主，有利於國家，設學頒訓者，既耶穌會也。(Mei and Tan 2017, p. 170)

At that time, the Jesuit Chinese mission was far too understaffed for seriously considering to establish colleges in China. However, as had been the case in Europe (see Casalini 2019), Vagnone and his fellow Jesuits may have been hoping that, at some point, benefactors would help to establish first elementary schools and then colleges placed under the authority of the Society even if most teachers would have been recruited among local scholars. Jesuits would have considered such an endeavor as an excellent way to foster local vocations as well. In other words, Vagnone's desire to see the excellence of his educational principles recognized by all may come with an ulterior motive: opening for his Order a way of establishing itself in China through the foundation of endowed schools—a strategy that had remarkably succeeded in Europe. The absence of a critical mass of local Christians and thus of potential benefactors was making such a project a mere dream, but Vagnone may have thought that he was thus preparing for the Society's future endeavors.[14]

## 5. Conclusions

Vagnone was focused on crafting spaces of agreement between Confucianism and a streak of Christianity that emphasizes the continuity between the realms of Nature and Grace while giving a prominent role to humanistic learning in the shaping of one's consciousness and capacity for action. He does not seem to have been as concerned as Ricci was with the drawing of a firm line among different Confucian schools: the *Tongyou jiaoyu* was to be read not as a competitor to the *Xiaoxue* but rather as embellishing and furthering its teachings.

Agreement among people religions and schools was more easily reached, in Vagnone's view, when focusing on questions as elemental as the love for one's children and the education to provide them with. A number of questions he discusses (age of marriage and conformity between the spouses, wet nursing, criteria to apply when it comes to child punishment) testify to his immersion in Chinese society.[15] The gradual and gentle approach of Vagnone is indeed that of a born educator. He builds bridges and points of convergence through the workings of a prose filled with reminiscences of natural and human realities.

The "eloquent rhetoric" of which Han Lin credits the book is rooted in a variety of simile and examples that resonates with Song-Ming Confucianism's naturalism and that attempts at the same time to progressively lead the reader from the realm of Nature to the one of Grace and divine realities. For Vagnone, there are dwelling places in the divine household for animals, sages and saints, and there are ways to circulate from one of these dwellings to another.

**Funding:** This research received no external funding.

**Institutional Review Board Statement:** Not applicable.

**Informed Consent Statement:** Not applicable.

**Data Availability Statement:** Not applicable.

**Conflicts of Interest:** The author declares no conflict of interest.

## Notes

[1] In what follows, references to the book will be given according to both the Chinese edition annotated by Thierry Meynard (Mei Qianli) and Tan Jie (Mei and Tan 2017) and the Chinese–English edition and translation offered by Giulia Falato (Falato 2020). In addition to the original of Vagnone's opus and an extremely careful English translation, Falato (2020) offers precious indications on the sources and the composition of the text as well as a wealth of lexical and historical information. In a slightly earlier publication, Thierry Meynard and Tan Jie provide the reader with a Chinese-language edition and presentation of the same text and insert it into the corpus of Late Ming Jesuit publications. My study draws extensively on these publications. With respect to the existing literature, the goal I specifically assign to myself is defined in the Introduction.

[2] Han Lin 韓霖was a Ming Dynasty book collector and scholar who also went by the names of Yu Gong 雨公 and Yu An 寓庵. He was born in the town of Chengguan in Xinjiang County. In 1621, during the reign of the Xizong Emperor (also known as Tianqi), he passed the imperial examination at the *juren* 举人 level.

[3] The term applies first to a children's book authored by the Tang dynasty scholar 李瀚.

[4] Among the Confucian classics, *The Great Learning* (*DaXue* 大學) emphasizes the cultivation of moral character, family education, national governance, and world peace as prerequisites for achieving "the great unity". The *Zhongyong* (中庸) posits that "peace through rituals and music" constitutes the cornerstone of good governance and social order, and it underscores the importance of the ritual system in establishing a harmonious and stable society.

[5] *Cifu* is a literary form of poetry in traditional Chinese culture, usually referring to a poem in the style of *Fu* 赋, a form of prose writing, combined with Ci 词, a type of lyric poetry, which was popular in the Tang and Song Dynasties.

[6] This expression is a latter-day reformulation of an idea first found in the conversations of Zhu Xi (Zhu 2014) related to the *Yijing* 易經 when he speaks of "discussing the *Yijing* [so as to] clarify human affairs [論易明人事]." The stress is on the observation of natural laws so as to clarify the logic of events apparently happening at random.

[7] For an overview of Ricci's strategy and further developments, see (Tan 2014).

[8] On Longobardi's objections to Ricci's approach, his reasons for not finding cultural equivalents between Christian doctrine and Confucianism, and his subsequent insistence on transliterating key Catholic terms, see Golden (2009).

[9] Let us note however that the term "philosophy' is also transliterated. This is the radical difference between "philosophy" and "theology", on the one hand, and the disciplines included in the Chinese system of knowledge that is implied by such transliterations.

[10] We refer here to the text of the *Tianzhu shiyi* available on the site https://ctext.org (Ricci n.d.) (accessed on 11 April 2023). We also consulted (Zhu 2001).

[11] My translation.

[12] See the example of Phocion entrusting his son to the care of a teacher (Falato 2020, pp. 160–61).

[13] The paragraph of the *Tongyou jiaoyu* that follows the one just quoted stresses the fact that, besides teaching about human and divine matters, Jesuits schools in Europe instruct children to respect the country's sovereign and foster filial piety. This may work as further reassurance for potential benefactors of such establishments in China. At the same time, in several places of the book, Vagnone writes that only the love and respect of God can ground respect for one's sovereign and one's parents.

[14] After 1843, the Jesuits would indeed open colleges in the Jiangnan region, and education (primary schools related to parishes, colleges, universities) would become an apostolic priority of the Second Jesuit Mission.

[15] Vagnone was first stationed in Nanjing and (for a brief time at the time of the publication of the *Tongyou jiaoyu*) in Shanxi, where his apostolate was remarkably successful. He prepared the book manuscript (as well as other writings) during his eight years exile in Macao (which may explain a number of references included in the book, as he could use the resources of the Jesuit college here). However, the book was finalized after his arrival in Jiangzhou 绛州 (today Xinjiang county, Shanxi province) in 1624.

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
