# Peer review of "Animals, Sages and Saints: Alfonso Vagnone’s Rhetorical Strategy in Chinese Context"

_religions, doi:10.3390/rel14060702_

Round 1
Reviewer 1 Report
This paper is a welcome contribution to the scholarship, although not entirely original, as it was touched upon by Falato and Meynard's previous studies on Tongyou jiaoyu. The author partially acknowledge these works, but mostly for the translation (Falato) and the annotated edition (Meynard). There are a few more points touched by the author, which have also been discussed in the previous scholarship on TYJY and should therefore be referenced. A few examples below:
1. p. 4: "[...] Nature to be the best teacher" also stated by Meynard
2. p.4 localisation of the nightingale also stated by Falato 2020
3. Vagnone's contribution to the Terms controversy was also discussed by Falato 2020 and in an article published in 2017 by Duan Chunsheng 高一志的辯è·ï¼šDeusæ¼¢è¯ä¹‹äº‰æ–°è€ƒ
Falato's study on Tongyou jiaoyu is also missing from the final bibliography, albeit being extensively cited in the article.
In addition, the abstract is too descriptive and should include one or two sentences stating the purpose of the study. The theoretical framework is sound, but a reader also expects to have a sense of the study's originality and contribution to the scholarship.
The scope of the paper appears more clearly in the introduction, although I believe that it shouldn't be only limited to the use of "animal analogies" (p.2), but also expand to how the rhetorical deployment of figures from the Western classical and religious literature (sages and saints) fit Vagnone's writing purpose.
Overall, the article is very well conceived and persuasively argued, convincingly demonstrating how Vagnone's rhetoric strove to highlight universally shared values between the Classical and Renaissance pedagogy and the Song-Ming Naturalism (unlike Ricci, who always favoured the pre-imperial version of Confucianism as a tool to spread the Catholic faith). It also provides some insightful points, such as the examination of Vagnone's use of some Chinese terms and the differences between Vagnone and Ricci's approach to Confucianism/Neo-Confucianism. When discussing animal parables/analogies, perhaps a few more cross-cultural examples directly quoted from European (e.g.Aesopian fables, bestiaries, etc.) and Chinese sources (e.g.leishu, etc.) would have further strengthened the argument. Li, Sher-shiueh is an excellent source on this topic.
From a stylistic perspective, there are a few minor inconsistencies which a final proofreading job would help solving. One example is the quotation "eloquent rhetoric" (pp.1 and 2) from Han Lin's foreword, which becomes "excellent rhetoric" on p. 10.
Author Response
Manuscript religions-2384006
Animals, Sages and Saints: Alfonso Vagnone’s Rhetorical Strategy in Chinese Context
My first draft did not give an explicit enough acknowledgment of Falato’s and Meynard’s contributions, without which the undertaking of this study would simply not have been possible, and I feel sorry for this. Besides specifying a few references throughout the article, I have inserted the following at the end of Note 1:
“Besides the original of Vagnone’s opus and an extremely careful English translation, Falato (2020) offers precious indications on the sources and the composition of the text as well as a wealth of lexical and historical information. In a slightly earlier publication, Thierry Meynard and Tan Jie provide the reader with a Chinese-language edition and presentation of the same text and insert it into the corpus of Late Ming Jesuit publications. My study draws extensively on these publications. With respect to the existing literature, the goal I specifically assign to myself is defined in the Introduction.”
- 4: "[...] Nature to be the best teacher" also stated by Meynard:
reference added
- 4 localisation of the nightingale also stated by Falato 2020:
refeence added
- Vagnone's contribution to the Terms controversy was also discussed by Falato 2020 and in an article published in 2017 by Duan Chunsheng 高一志的辯è·ï¼šDeusæ¼¢è¯ä¹‹äº‰æ–°è€ƒ:
references added
Falato's study on Tongyou jiaoyu is also missing from the final bibliography, albeit being extensively cited in the article:
I am sorry I missed this basic reference when establishing my bibliography. Issue solved.
In addition, the abstract is too descriptive and should include one or two sentences stating the purpose of the study. The theoretical framework is sound, but a reader also expects to have a sense of the study's originality and contribution to the scholarship:
See the following change:
“This is done through various rhetorical devices, one of them being the extensive use of animal simile, which prepares considerations about behavioral models to be found in (Pagan) Sages and (Christian) saints. This study focuses on the rhetorical and narrative methods through which Vagnone grounds a gradation and continuum between Nature and Grace, inserting his pedagogical considerations into a carefully crafted apologetics.”
The scope of the paper appears more clearly in the introduction, although I believe that it shouldn't be only limited to the use of "animal analogies" (p.2), but also expand to how the rhetorical deployment of figures from the Western classical and religious literature (sages and saints) fit Vagnone's writing purpose.
See following change;
“I pay special attention to three aspects of Vagnone's rhetoric: first, his use of animal analogies; second, the way these analogies are mobilized into a continuum that is both pedagogical and theological; third his implicit openness to the whole of the Confucian tradition, rather than merely one particular school within it. By examining these particular features I intend to highlight etc “
From a stylistic perspective, there are a few minor inconsistencies which a final proofreading job would help solving. One example is the quotation "eloquent rhetoric" (pp.1 and 2) from Han Lin's foreword, which becomes "excellent rhetoric" on p. 10.
Issues hopefully solved.

Reviewer 2 Report
This is one of the most interesting papers I have reviewed in some time. For the most part, it is a model of clarity and good organization. Better still, the topic is fascinating.
There are, however, a few typos and hard-to-understand passages. In addition, I have one substantive comment.
p. 2: present be presented
p. 2: and argue [argument?]
p. 4: raccoon the dog [missing comma]
p. 4: Vagnone links into one streak the Western humanistic tradition with the propensity: This does NOT make sense; must rewrite/explain better.
p. 5: or yet Alexander--> the use of "yet" makes no sense; is this a typo, or is something missing here?
My only substantive comment centers on the references to the Terms Controversy in relation to Japan, which is first briefly mentioned on page 6, followed by an interesting summation of the implications of the controversy (i.e., "it looks as if finding a Chinese equivalent for the sacred science that Theology is meant to be would have run the risk to corrupt the purity of the doctrine it conveys.") That summation is interesting but entirely insufficient in this context. The author/s MUST briefly explain this controversy, in perhaps 100 words or so: what was it and what consequences did it have? This is especially important because the paper raises the issue again later on.
Author Response
Manuscript religions-2384006:
Animals, Sages and Saints: Alfonso Vagnone’s Rhetorical Strategy in Chinese Context
Thanks for pointing out a number of typos and stylistic issues. I think I have taken care of all of them.
My only substantive comment centers on the references to the Terms Controversy in relation to Japan, which is first briefly mentioned on page 6, followed by an interesting summation of the implications of the controversy (i.e., "it looks as if finding a Chinese equivalent for the sacred science that Theology is meant to be would have run the risk to corrupt the purity of the doctrine it conveys.") That summation is interesting but entirely insufficient in this context. The author/s MUST briefly explain this controversy, in perhaps 100 words or so: what was it and what consequences did it have? This is especially important because the paper raises the issue again later on.
I have inserted the following paragraph:
“Influenced by Francis Xavier’s way of preaching, the Society of Jesus in Japan had made theological and linguistic choices that were far less accommodating than those of Ricci, and had transliterated the name of God from the Latin Deus. Besides, the end of the sixteenth century and the beginning of the following century were marked by a more pessimistic theology than that which Ricci had enjoyed during his training. After Ricci had died, almost all the Jesuits in the province of Japan and some of the Jesuits in the vice-province of China protested, often violently, against the linguistic and ritual accommodations already made, and even threatened to appeal to the Inquisition, based in Manila. The majority of the Jesuits of the China mission, led by Nicolas Trigault (1577-1628) and Alfonso Vagnone, nevertheless were remaining faithful to the accommodations decided by the founder of the mission (Duan 2017; Falato 2020, p.57-58). After the conference of Jiading (near Shanghai) which, starting in December 1627, brought together eleven of the Jesuits involved in the controversy, the Visitor, André Palmeiro (1569-1635) decided to authorize the use of the term Tianzhu (Lord of Heaven) to designate the Christian God, but to banish those of Shangdi (Lord of Above) and Tian (Heaven) which Ricci was also using. On the other hand, the lawfulness of attending rites in honor of Confucius and the ancestors was confirmed (see Pina 2003; Kim 2004; Brockney 2014).”
